# A Mixed-Method Evaluation of a Rural Elementary School Implementing the Coordinated Approach to Child Health (CATCH) Program

**DOI:** 10.3390/nu15122729

**Published:** 2023-06-13

**Authors:** Carmen D. Samuel-Hodge, Ziya Gizlice, Alexis R. Guy, Kathryn Bernstein, Aurore Y. Victor, Tyler George, Trevor S. Hamlett, Lisa M. Harrison

**Affiliations:** 1Gillings School of Global Public Health, Department of Nutrition, Center for Promotion and Disease Prevention, University of North Carolina at Chapel Hill, 1700 Martin Luther King Jr. Blvd., Room 216, CB #7426, Chapel Hill, NC 27599-7426, USA; 2Center for Health Promotion & Disease Prevention, University of North Carolina at Chapel Hill, 1700 Martin Luther King Jr. Blvd., CB #7426, Chapel Hill, NC 27599-7426, USA; zgizlice@gmail.com (Z.G.); alexisrachelguy@gmail.com (A.R.G.); kathryn.bernstein@gmail.com (K.B.); ayvictor1@gmail.com (A.Y.V.); trevorhamlett721@gmail.com (T.S.H.); 3Division of General Medicine, School of Medicine, University of North Carolina at Chapel Hill, 102 Mason Farm Rd. #3100, Chapel Hill, NC 27599, USA; tyler_george@med.unc.edu; 4Granville Vance Public Health, 1032 College St., Oxford, NC 27565, USA; lharrison@gvdhd.org

**Keywords:** rural populations, health promotion, childhood obesity, overweight prevention, body mass index, physical fitness, school-based intervention, qualitative data

## Abstract

Despite children living in rural US areas having 26% greater odds of being affected by obesity compared to those living in urban areas, the implementation of evidence-based programs in rural schools is rare. We collected quantitative data (weight and height) from 272 racially and ethnically diverse students at baseline, and qualitative data from students (4 focus groups), parents, and school staff (16 semi-structured interviews and 29 surveys) to evaluate program outcomes and perceptions. At the 2-year follow-up, paired data from 157 students, represented by racial/ethnic groups of 59% non-Hispanic White, 31% non-Hispanic Black, and 10% Hispanic, showed an overall mean change (SD) in BMI z-score of −0.04 (0.59), a decrease of −0.08 (0.69) in boys, and a significant −0.18 (0.33) decrease among Hispanic students. Boys had a mean decrease in obesity prevalence of 3 percentage points (from 17% to 14%), and Hispanic students had the largest mean decrease in BMI percentile. Qualitative data showed positive perceptions of the CATCH program and its implementation. This community-engaged research, with collaboration from an academic institution, a health department, a local wellness coalition, and a rural elementary school, demonstrated successful CATCH program implementation and showed promising outcomes in mean BMI changes.

## 1. Introduction

In the United States, childhood obesity, defined as having a body mass index (BMI) at the ≥95th percentile for age and sex [1], remains a significant public health concern, with 2017–2020 data showing an obesity prevalence of 19.7% among children and adolescents aged 2–19 years [2]. Obesity prevalence by age group was 12.7% among children 2–5 years of age, 20.7% among those 6–11 years of age, and 22.2% among those 12–19 years of age [1,2]. These high rates are concerning because childhood obesity is known to increase the risk for a number of adverse chronic conditions, including hypertension, hypercholesterolemia, type 2 diabetes, sleep apnea, and joint problems [1].

While the impact of childhood obesity is widespread, its burden is not evenly distributed [1]. Estimates from the National Health and Nutrition Examination Survey revealed that between 2013 and 2016, urbanization was inversely associated with child obesity, with higher rates in non-metropolitan areas [3]. Additionally, non-Hispanic Black and Hispanic children are much more likely to be affected by obesity than White and Asian children in all age categories [1]. The elementary school in this report is located in a mostly rural county in eastern North Carolina with a population of 64.2% non-Hispanic White, 31.8% non-Hispanic Black, and 9.5% Hispanic or Latino [4], which puts its school population at high risk for childhood obesity.

Although differences in definitions of rurality make comparisons challenging, there is good evidence to support its negative impact on weight status. A meta-analysis of five studies found that children living in rural areas have 26% greater odds of being affected by obesity compared to those living in urban areas [5]. Moreover, within rural populations, some groups, including non-Hispanic Black residents and those with lower educational achievement, are disproportionately affected [6].

These differences in the burden of childhood obesity point to high-risk rural, poor, and racial/ethnic minority populations as obvious targets for the implementation of evidence-based interventions, especially those designed for school settings, where there is the greatest potential to reach a large number of children. The Coordinated Approach to Child Health (CATCH) program [7,8] is one such evidence-based intervention with well-documented effectiveness in various school settings [7,9,10,11]. CATCH has proven to be effective in improving dietary and physical activity behaviors and reducing the prevalence of overweight and obesity [12,13,14]. Since the built environment and school nutrition and physical activity policies are implicated in the development of childhood overweight and obesity [15,16,17], they represent good targets for obesity prevention interventions such as CATCH [18]. The CATCH program provides children with the knowledge and environment to support healthy eating and physical activity by bringing together four key program components (classroom curriculum and policies, child nutrition services, physical education, and family involvement) [7]. By addressing nutrition and physical activity-related policies, CATCH has the potential to lessen the disparate burden of childhood obesity observed in rural school-age children.

Unfortunately, the potential public health impact of CATCH implementation in rural school settings is not being realized. To date, there are very few published reports on CATCH implementation in rural school settings [15,16,17,19], and only one to our knowledge in rural southeastern states, where the rates of obesity are particularly high [19]. Among the few reports of CATCH implementation in rural settings, racial minority groups are not well represented in the sample populations. This descriptive paper reports findings from an academic community-engaged research study [20] evaluating the implementation of the CATCH program in a rural North Carolina elementary school with a racially and ethnically diverse student population. We report here the weight status and physical fitness changes after two years of CATCH program implementation, along with qualitative data describing staff, parent, and student perceptions of the CATCH program.

## 2. Materials and Methods

Intervention (CATCH) Implementation: Implementation of the CATCH program began in March 2015 with the training of school staff. The school leadership decided that the most feasible approach to implementing CATCH program components in this rural school with limited resources was to use a sequenced approach rather than attempting to implement all program components at once for all students. Among the four core program components (classroom curriculum and policies, child nutrition services, physical education, and family involvement), the school began implementation at the start of the 2015–2016 school year by first targeting classroom curriculum and policies, and physical education. Students in kindergarten and first grade were the first to receive classroom curriculum content, but all grades received some lessons, and all students were exposed to school-wide announcements with CATCH messaging (“Go, Slow, Whoa, (GSW),” MVP (Moves and stays active, Values healthy eating and mindsets, Practices healthy behaviors), etc.). At the time of our baseline anthropometric measurements (April 2016), there was limited implementation of the classroom curricula in third- to fifth-grade classes; this increased starting in August 2017, and we collected follow-up measurements in April 2018.

Quantitative Data: Study data included a combination of measures taken by the study staff, and data collected and provided by the school administration. Study staff collected weight and height, while the school provided demographic and fitness data (Progressive Aerobic Cardiovascular Endurance Run (PACER) test [21] and sit-ups). Trained study staff collected weight measurements using an electronic scale (Seca model 874; Seca Columbia, MD, USA). Two weights taken without shoes were obtained and averaged; if the difference between the two weight values was greater than 1 lb, a third weight was obtained, and the three weights were averaged. We used a portable stadiometer (Shorr Productions, Olney, MD, USA) to obtain the average of two height measures (with less than ¼ inch difference) taken without shoes. For calculations of the body mass index (BMI) percentiles, age was calculated using date of birth and the measurement date.

Qualitative Data: Qualitative data collection included an online survey of school staff, semi-structured interviews with parents and staff, and focus groups with third- to fifth-grade students. All qualitative data were collected between May and December 2017. Adult participants provided informed consent, and children provided written assent along with passive parental permission or consent. Adult participants received a USD 20 incentive for survey or interview completion; students received small incentives such as stickers, colored pencils, erasers, pencil cases, and reusable snack bags. The online survey was sent to school staff to assess their perceptions of the CATCH program as an innovation (advantages, likely outcomes, and compatibility with the school environment), its implementation, and the staff’s commitment to the school implementing the CATCH program. The survey included items adapted from subscales of validated surveys measuring constructs of (1) innovation perceptions (relative advantage and compatibility of the program with the organization) [22], and two types of commitment to organizational change (affective commitment or identification with and involvement in the organization, and normative commitment or perceived obligation to the organization) [23]. Trained research staff conducted semi-structured interviews with the parents/caregivers of third- to fifth-graders and school staff, using an interview guide with items that gathered demographic information and their perceptions of the CATCH program being implemented at the school. Similarly, trained moderators conducted focus groups with small groups of third- to fifth-grade students, to gather their views of the CATCH program (what they knew about CATCH and how they felt about the changes that came with CATCH implementation). Interviews and focus groups were audio-recorded, and digital files were later transcribed verbatim. We created a coding dictionary with codes and definitions for the focus groups and interviews and refined our coding system after two reviewers independently coded the first transcript. All transcripts were then independently coded by two coders who later resolved any coding discrepancies through consensus. Our analytic approach involved conventional content analysis, where coding categories were derived directly from the transcribed text [24].

Statistical Analysis: Baseline sample characteristics and survey data were summarized using descriptive statistics, such as means, percentages, standard deviations, etc. Analyses of changes in BMI status and fitness were conducted using paired t-tests. All analyses were conducted using SAS Version 9.4 (SAS Institute, Cary, NC, USA).

## 3. Results

Quantitative: Table 1 presents the demographic characteristics of students who provided anthropometric measurements at both time points. At baseline measurement, a total of 349 students (Pre-K to fifth grade) were enrolled, and we obtained data from 272 in kindergarten to fourth grade. Table 2 shows the baseline relationships, with BMI percentile and measures of physical fitness and academic performance from the third- to fifth-graders. Higher BMI percentiles were significantly associated with lower physical fitness, as measured by the PACER test and sit-ups, and lower reading and math scale scores. Higher reading scale scores were significantly associated with better physical fitness scores, whereas math scale scores were not.

At the 2018 follow-up, 216 of the 272 students measured at baseline were still enrolled at the school and 157 (73%) provided data. The sample included 86 girls (55%) and 71 boys (45%), represented in racial/ethnic groups of 59% non-Hispanic White, 31% non-Hispanic Black, and 10% Hispanic. The proportions of students categorized with overweight and obesity at baseline are shown in Table 3. The proportions with overweight and obesity by BMI category did not differ significantly by sex, racial/ethnic subgroup, or grade. Although statistically non-significant, more girls (20%) than boys (17%) were situated in the obesity category, and obesity rates were lowest among students identified as non-Hispanic White, and highest among those identified as non-Hispanic Black. Overall, we observed a mean decrease in the proportion of students in the obesity category of 3 percentage points (17% to 14%) among boys and an increase of 2% among girls (20% to 22%), between 2016 and 2018.

Figure 1 shows paired mean changes in BMI z-scores in the overall sample and by sex, race/ethnicity, and grade. Figure 2 shows changes in BMI percentile in the overall sample and by sex, race/ethnicity, and grade. Only overall changes are provided for physical fitness in Figure 3. Overall, there was a −0.04 (SD 0.59, *p* = 0.35) decrease in the mean BMI z-score. Boys showed a larger decrease in the mean BMI z-score (−0.08, SD 0.69, *p* = 0.31) compared to girls. By race/ethnicity, only Hispanic students showed a significant decrease, of −0.18 (SD 0.33, *p* < 0.05), in mean BMI z-scores. Although statistically non-significant, students in third grade at follow-up had the largest decrease, of −0.08 (SD 0.59, *p* = 0.41), in mean BMI z-scores. Overall, the mean BMI percentile was lower by 1.7 percentile points (*p* = 0.19). By gender, there was a bigger decrease seen among boys than girls (a −2.5 change in boys (*p* = 0.28) vs. a −1.1 change in girls (*p* = 0.48)). Comparisons among racial/ethnic categories showed that Hispanic students had the biggest decrease in BMI percentile, of −5.05 percentile points (*p* = 0.09), with a smaller decrease among non-Hispanic White students, and no change among non-Hispanic Black students.

In addition to BMI data collected by the study staff, fitness data collected by the school staff were also provided for 2017 and 2018 (Figure 3). Fitness level data measured by the PACER test, push-ups, and sit-ups showed some improvement at follow-up. Comparisons using paired data from the subset of participants with measures at both time points showed a significant (*p* < 0.0001) decrease in PACER test scores from 2017 (mean 20.5, SD 11.7) to 2018 (mean 15.3, SD 9.0). Changes in fitness measured by push-ups showed no significant change over time (2017 mean of 7.3, SD 6.5; 2018 mean of 7.2, SD 6.5). Fitness measured by sit-ups, however, showed significant (*p* < 0.0001) improvement, with an increase in mean scores of 17.2 (SD 5.8) in 2017 to 25.1 (SD 8.8) in 2018.

Qualitative: Table 4 shows the demographics and perceptions of 29 out of approximately 40 staff members who responded to the online survey. Respondents were mostly female (86%) and teachers (83%), with school tenures of 1 to 27 years (median of 8 years), and most (54%) had completed training in CATCH delivery. Most respondents (85%) felt that their knowledge of CATCH was medium to high, and a majority reported observing several CATCH program components being implemented (e.g., class lessons and activities, schoolwide CATCH messaging, and changes in school lunch and physical activity). Family and community engagement in CATCH were less often observed. CATCH outcomes viewed as “very likely” by most respondents included improvements in physical fitness and eating habits. While most outcomes were rated very or somewhat likely, the least likely outcomes included fewer days missed from school and a reduction in overweight/obesity. Measures of the relative advantage and compatibility of CATCH in the school showed high mean scores for both, with higher scores for compatibility (4.0 and 4.6 respectively). The high advantage scores indicated that staff viewed CATCH as advantageous for the school and having the potential to positively affect childhood obesity; high compatibility scores meant that staff perceived CATCH as fitting well with the school’s health and wellness activities. Likewise, both measures of commitment to organizational change (e.g., adopting CATCH) were also high, with higher scores for affective commitment, or believing in the value of the changes that would come with the school’s adoption of the CATCH program.

Our qualitative findings from the focus groups with third- to fifth-grade students, and interviews with staff and parents/caregivers, are summarized below and in Table 5 (see Appendix A Appendix A for more details.)

Focus Groups: Students generally recognized that the CATCH program emphasizes healthy foods and physical activity (PA). They identified significant messaging about a key component of CATCH—“Go, Slow, Whoa (GSW)” foods—throughout the school and were able to broadly explain the differences among these food categories. Overall, understanding GSW foods influenced students to try new foods and select healthier food options, although a few students made no changes. As the “boss” of CATCH, students would increase healthy and decrease unhealthy food options, and they mostly expressed a willingness to discuss CATCH with their parents to impact their parents’ habits. Students would also increase PA in schools and incorporate rewards to support this. Consistent with these views of PA, students expressed disappointment with the limited opportunities for physical activity during recess, physical education, and/or “Fit Fridays.” Similarly, while students generally welcomed “brain breaks”/”energizers” (brain breaks or classroom energizers are physical activity breaks used to help kids re-focus and reactivate their brains after they’ve been sitting for extended periods of time) and class GSW lessons, some preferred energizers with more PA or dancing, and sought more in-depth discussions of GSW within classes.

Parent/Caregiver Interviews: While some parents had a general understanding of CATCH program goals, others were unfamiliar with the CATCH program specifics. Overall, however, parents recognized several school changes that occurred because of CATCH, including: (1) increased variety in activities and equipment for PE and recess; (2) discussion in class lessons of GSW and a healthy lifestyle; (3) increased healthy food options on the school menu; (4) increased activity and focus for children during “brain breaks”; and (5) use of non-food rewards by teachers. Consistent with the students’ views, parents noted that many of their children understood GSW foods (although some children with disabilities had challenges with GSW) and that some of their children were more willing to try new foods and made healthier food choices. Parents also expressed that their children encouraged them to make healthier choices at home. CATCH also helped to reinforce and/or increase PA for some families. Finally, parents recommended expanding the CATCH program to other schools, and sought better communication about CATCH from school staff and greater family participation in CATCH activities.

Staff Interviews: In general, staff were familiar with CATCH goals and key program components. Similar to students and parents, staff observed students discussing GSW, although some younger students misunderstood GSW, and making efforts to change their food choices at school. Staff likewise understood GSW and had healthy food availability at teacher meetings; some teachers were also more active at school after CATCH. When asked directly, staff viewed brain breaks, nutrition lessons, and changes to the school lunch offerings as most important. Staff felt that brain breaks improved the focus and engagement of students in class. While some staff easily implemented CATCH nutrition concepts within their lessons, others would welcome more guidance. Areas where staff felt that improvements were needed included having more time for staff to collaborate with each other and share strategies to implement CATCH more effectively in the classroom, and increasinghealthy food options at school to help model CATCH concepts of GSW.

## 4. Discussion

This evaluation study represents a collaborative effort between an academic institution, a wellness coalition supported by an academic health department, and a rural elementary school, to describe and document the implementation of CATCH in a rural, racially diverse population of children in the southeastern US. We engaged and mobilized students from the academic institution, and community members from the wellness coalition, local health department, and the elementary school to participate in collecting both quantitative and qualitative data that helped us better describe our starting point and assess changes over time. This community-engaged research approach [20] not only made the research feasible, it also strengthened the community’s capacity to address a health-related problem through a research partnership to implement an evidence-based intervention (CATCH). One of the important adaptations made to implement the CATCH program was the school’s decision to use an incremental approach in the grades receiving the program and the CATCH program components offered (e.g., starting with what was viewed as most important and manageable and then expanding over time). Although this approach undoubtedly affected the program dose received by students, the pacing may have benefited program implementation by reducing the staff burden and allowing more time to make the program fit the school-specific context. These adaptations made by the school leadership in implementing CATCH were consistent with key domains of the Consolidated Framework for Implementation Research (CFIR) [25] (e.g., adapting an intervention to fit the school or inner setting, and the community or outer setting).

Our evaluation study began with measuring the BMI status of students at the school and the larger community because the only available obesity data for the county was based on a sample of 60 children, 5–11 years of age, seen by the WIC program, child health clinics, and school-based health centers in 2015 [26]. In 2016, our team measured weights and heights in over 4200 pre-kindergarten and elementary school children in Granville and Vance counties and observed obesity rates of 24.9% overall, with higher rates among Hispanic children (29.9%) and non-Hispanic Black children (26%) (see Appendix A Appendix A). In this elementary school, we observed at baseline an overall obesity rate of 18.8%, which was similar to the 2015 NC county-level rate of 18% among children 5–11 years of age [26], and the 2015–2016 national prevalence of 18.8% in children 6–8 years, and 18.5% in those 9–11 years old [27]. Our obesity rate of 23.5% among non-Hispanic Black students was higher than both county and national rates. Among the 157 students measured at both time points, the overall obesity prevalence of 18.5% was also similar to county and national rates.

Given the level of research funding and our efforts to keep a low burden of data collection on the part of the school, we collected only baseline and follow-up weight and height measures to calculate changes in BMI status, and the school provided data from three tests of physical fitness. Our BMI z-score changes compared favorably with findings from the only other identified implementation of CATCH in a southeastern US sample of elementary school children [19], which found at 7-month follow-up an increase of 0.05 in the mean BMI z-score (SD 0.42, *p* = 0.28) compared to our 2-year mean decrease of 0.04 overall, with 0.08 among boys, and a significant decrease of 0.18 among Hispanic students. The 2023 Community Guide systematic review of school dietary and physical activity interventions (*n* = 24 studies), found that in 10 of 12 studies where the BMI z-score could be combined to calculate a median effect, the median decrease in BMI z-score was 0.07 [28]. Our median change in BMI z-score was 0.05. This recent systematic review also reported a median decrease in the prevalence of overweight and obesity of 2.5 percentage points [28]; we observed among boys only, a 3.0% mean decrease in obesity prevalence. An important point to make here is one made in the recent Community Guide systematic review on how they viewed favorable study outcomes: given the national trends of modest increases in childhood obesity prevalence, the review viewed “studies without a control group that reported a decrease or no change in weight-related outcomes as favorable, regardless of statistical significance, because this showed potential for a decreased rate of change in BMI z-score, overweight, or obesity prevalence” [28].

Our outcomes in physical fitness were mixed, with a significant increase in sit-ups, a significant decrease in PACER test scores, and no change in push-ups between 2016 and 2018. Since all physical fitness data were collected by school staff, we can only speculate as to why, for example, there was an unexpected decrease in the PACER test scores. Compared to measuring sit-ups and push-ups, the PACER test may have been more difficult to score. One possible reason for a lower posttest mean score could be measurement error due to non-adherence to scoring protocols (e.g., not having a practice day before administering the test, scoring more than two to three students at a time, or younger students not understanding the instructions) [21]. Although we did not collect data on student physical activity behaviors, a recent study in elementary school children has shown performance on physical fitness tests, including the three used in this study, was significantly associated with minutes of physical activity during physical education classes and recess [29]. Moreover, we observed at baseline, significant positive associations between physical fitness test scores and reading but not math scale scores. Other studies have found similar associations. Among fourth-, sixth-, and eighth-grade public school students, one study found statistically significant relationships between the number of fitness tests passed and passing both math and English state-level tests [30]. More broadly, there is moderate evidence from systematic reviews supporting the association between physical activity, cognitive function, and performance on academic achievement tests in children [31,32]. Related to physical activity, our qualitative findings highlight the generally positive views of the changes in physical education, physical activity, and brain breaks or classroom energizers that came with implementing the CATCH program. There is now increased interest in brain breaks and other classroom-based physical activities because of their potential to improve cognitive function, which is a predictor of academic achievement in children [33].

To convey a more complete story of CATCH program implementation in this small rural elementary school, we qualitatively gathered the views of staff, students, and parents/caregivers. The survey with school staff provided insights into how they “saw” CATCH being implemented, their thoughts about expected outcomes, and their commitment to the school in its adoption of CATCH. The overall favorable view of the CATCH program and its implementation was likely due to the team-based approach used at the school in planning for its implementation, and the ongoing communication between the school principal and the research partners at the academic health department, which also served as the backbone organization for the local wellness coalition. The perspectives of third- to fifth-graders in the focus groups suggest that they understood the overall message of “Go, Slow, Whoa” as a guide to making healthy food choices, and that the CATCH program was impacting their dietary and physical activity behaviors. Since parents of third- to fifth-grade students were not directly targeted in the implementation of CATCH because of the priority given to other program components, their views on wanting to know more about the program and becoming more engaged in school activities are perfectly understandable. Despite limited engagement and communication between the school and families, parents knew about CATCH program activities and information because their children shared this information with them.

This mixed-method evaluation study conducted as community-engaged research helps to fill the gap in the implementation of evidence-based interventions such as CATCH in rural school settings with racially and ethnically diverse populations, such as those in the southeastern US. This collaborative and adaptive approach to implementing evidence-based interventions in low-resourced public school systems may make it more feasible for small rural schools to benefit from having programs with demonstrated effectiveness delivered in their settings. This small evaluation study also demonstrated that adaptations to reduce the complexity of program delivery should not be seen mainly as a threat to delivery fidelity, but more as an opportunity to improve the compatibility of the program with the context in which it will be implemented. In this study, we delayed the measurement of program outcomes to better align with how the program was implemented; this approach allowed us to share a more positive story of program effectiveness with the school and other community members. In conducting this study, we also acknowledge that there were noteworthy limitations. A major limitation to attributing the observed BMI outcomes to the CATCH program was the lack of a control or comparison group. Given the normal changes in BMI for this age group of children, we used changes in BMI z-scores instead of BMI because BMI z-scores allowed us to make comparisons between boys and girls and children of different ages, thus accounting for growth patterns and genetic influences. Moreover, the observed mean decreases in BMI z-score and BMI percentile would not be expected given the upward trend in the prevalence of obesity among US children [27]. Other limitations include a small sample size, which limited sub-group analyses, limited outcome measures, particularly measures of student behavioral change, and the extent to which the CATCH program positively impacted academic performance. Schools must often choose between meeting metrics for student wellness and academic proficiency; demonstrating that programs can deliver both outcomes in under-resourced rural school settings would go a long way in garnering financial and other resources to support more broadscale adoption of programs such as CATCH.

## Figures and Tables

**Figure 1 nutrients-15-02729-f001:**
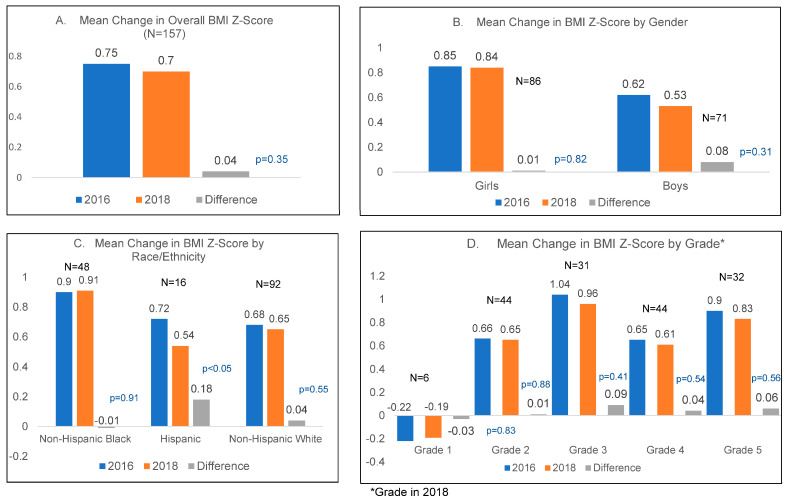
Mean change in BMI Z-scores.

**Figure 2 nutrients-15-02729-f002:**
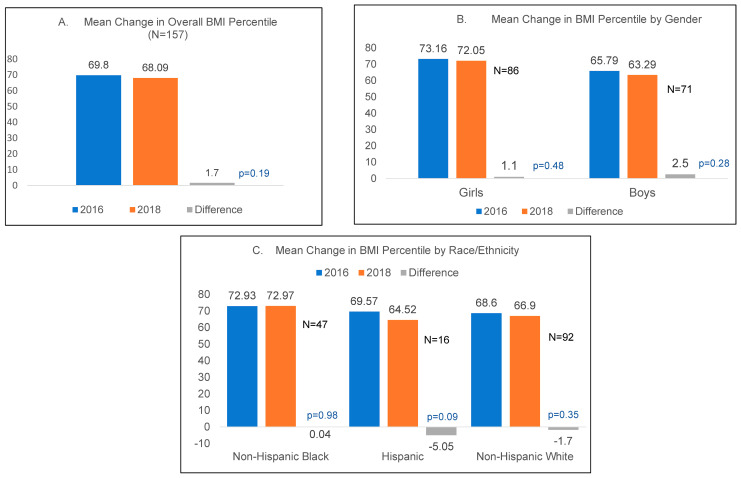
Mean change in BMI percentile.

**Figure 3 nutrients-15-02729-f003:**
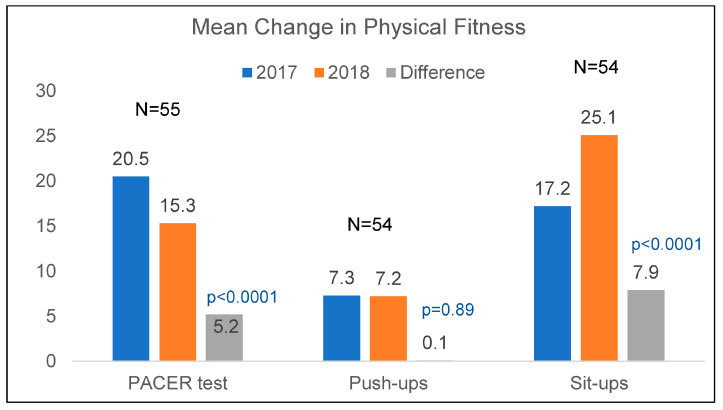
Mean change in overall physical fitness.

**Table 1 nutrients-15-02729-t001:** Percentage of children with overweight and obesity by demographic characteristics and grade (N = 272).

	No Overweight/Obesity * N (%)	Overweight N (%)	Obesity% N (%)	*p*-Value
Girls (N = 138)	83 (60.1)	55 (39.9)	27 (19.6)	0.61
Boys (N = 134)	88 (65.7)	46 (34.3)	24 (17.9)	-
Ethnicity				0.78
Non-Hispanic White (N = 154)	101 (65.6)	53 (34.4)	24 (15.6)	-
Non-Hispanic Black (N = 85)	51 (60.2)	34 (40)	20 (23.5)	-
Hispanic or Latino (N = 26)	15 (57.7)	11 (42.3)	5 (19.2)	-
Other ** (N = 7)	4 (57.1)	3 (42.9)	2 (28.6)	-
Grade				0.01
Kindergarten (N = 65)	51 (78.5)	14 (21.5)	10 (15.4)	-
1st grade (N = 48)	25 (52.1)	23 (47.9)	13 (27.1)	-
2nd grade (N = 57)	35 (61.4)	22 (38.6)	8 (14)	-
3rd grade (N = 46)	22 (47.8)	24 (52.2)	10 (21.7)	-
4th grade (N = 56)	38 (67.9)	18 (32.1)	10 (17.9)	-

* Includes children categorized as normal and underweight. The overweight category includes children with obesity. ** Includes children identified as mixed race, American Indian or Alaska Native, and Asian or Asian Pacific Islanders.

**Table 2 nutrients-15-02729-t002:** Baseline relationships among BMI percentile, physical fitness, and academic performance measures.

Variables	1	2	3	4	5	6
BMI percentile	-					
2. PACER test (*n* = 85)	−0.23 *	-				
3. Push-ups (*n* = 84)	−0.15	0.36 ***	-			
4. Sit-ups (*n* = 84)	−0.33 **	0.53 ^§^	0.47 ^§^	-		
5. Reading scale score (*n* = 85)	−0.41 ^§^	0.20 *	0.33 ***	0.36 ***	-	
6. Math scale score (*n* = 69)	−0.32 **	0.06	0.09	0.14	0.56 ^§^	-
Variables	1	2	3	4	5	6
Mean (SD)	69.0 (26.6)	22.8 (13.5)	8.5 (7.6)	21.1 (9.2)	437.5 (10.8)	446.2 (10.2)
Median	75.5	20.0	8.0	20.0	437.0	445.0

Pearson correlation coefficients; Fitness and reading scores (3rd–5th grades); Math scores (4th–5th grades only). * *p* < 0.05; ** *p* < 0.01; *** *p* < 0.001; ^§^
*p* < 0.0001.

**Table 3 nutrients-15-02729-t003:** Baseline BMI for age by demographic characteristics and grade for students providing BMI follow-up data (N = 157).

	Normal/Underweight * N (%)	Overweight N (%)	Obesity N (%)	*p*-Value
Girls (N = 86)	50 (58)	19 (22)	17 (20)	0.57
Boys (N = 71)	47 (66)	12 (17)	12 (17)	-
Ethnicity				0.98
Non-Hispanic White	58 (63)	18 (20)	16 (17)	-
Non-Hispanic Black	29 (60)	9 (19)	10 (21)	-
Hispanic or Latino	9 (56)	4 (25)	3 (19)	-
Other	1 (100)	0 (0)	0 (0)	-
Grade				0.16
1st	6 (100)	0 (0)	0 (0)	-
2nd	30 (68)	5 (11)	9 (20)	-
3rd	16 (52)	8 (26)	7 (23)	-
4th	25 (66)	7 (16)	8 (18)	-
5th	16 (50)	11 (34)	5 (16)	

* Obesity = BMI ≥ 95th percentile; overweight = BMI ≥85th–<95th percentile; normal weight = 5th–<85th percentile (includes children categorized as underweight (<5th percentile)).

**Table 4 nutrients-15-02729-t004:** Staff survey results—perceptions of CATCH, schoolwide implementation, and commitment to organizational change.

Demographics and Role in CATCH Implementation (N = 29)			
	N (%)		
Gender—Female	25 (86.2)		
Years at School, mean (SD)	10.2 (8.1)		
Position at School			
Teacher	24 (82.8)		
Administrative or other staff	5 (17.2)		
Role in Planning and Implementation *	22 (53.7)		
CATCH training (participant)			
Planning committee member	5 (12.2)		
CATCH implementation team member	7 (17.1)		
Other role	4 (9.8)		
No role	3 (7.3)		
**Knowledge and Perceptions of CATCH Program Implementation (N = 27)**			
Knowledge of CATCH Program	High	Medium	Low
	5 (18.5)	18 (66.7)	4 (14.8)
Observed CATCH Program Components Implemented, N (%) of total respondents			
Classroom lessons and activities	23 (79.3)		
Changes in types of foods offered at lunch	15 (51.7)		
Changes in how active kids are during PE	18 (62.1)		
Children using the “Go, Slow, Whoa” food messaging	26 (89.6)		
Family involvement to support healthier eating and physical activity habits	12 (41.4)		
Community events to support healthier eating and physical activity habits in families	11 (37.9)		
Perceived likelihood of CATCH leading to outcomes for children, N (% of total item responses)	Very Likely	Somewhat Likely	Not Very Likely
Fewer children in overweight/obesity category	7 (25.9)	17 (63.0)	3 (11.1)
Students more physically fit	14 (51.8)	12 (44.4)	1 (3.7)
Students with healthier eating habits	17 (63.0)	10 (37.0)	0 (0)
Fewer days missed from school	5 (18.5)	18 (66.7)	4 (14.8)
Better academic performance	10 (37.0)	16 (59.3)	1(3.7)
**Innovation Perceptions and Commitment to Organizational Change, (N = 27)**			
Innovation (CATCH) Perceptions Subscale Scores, mean (SD) **			
CATCH Advantage (5 items)	4.0 (0.38)		
CATCH Compatibility (3 items)	4.6 (0.11)		
Overall score	4.2 (0.39)		
Commitment to Organizational Change Subscale Scores, mean (SD) **			
Affective Commitment	4.6 (0.12)		
Normative Commitment	4.0 (0.37)		
Overall score	4.3 (0.44)		

* Some staff had multiple roles. ** 5-point Likert scale of “disagree a lot” to “agree a lot”, with maximum item score = 5, minimum = 1; higher scores = better or more positive scores. Innovation perceptions (8 items total, with Advantage = 5 items and Compatibility = 3 items); Commitment to organizational change (12 items; 6 items per subscale). Items were adapted to include CATCH as the innovation.

**Table 5 nutrients-15-02729-t005:** Qualitative data from focus groups and semi-structured interviews.

Focus Groups—3rd to 5th Grade Students	
Domains	Themes and Illustrative Quotes
CATCH Knowledge	*“It’s about how you exercise and how you eat.”*
Recommendations if Students were the “Boss” of CATCH	Increase Physical Activity and Rewards: *“You could expand our PE time, physical education.”* *“Work competition where they have to exercise a lot and whoever wins gets like field day.”* Increase Healthy and Decrease Unhealthy Foods: “*A little bit more baked foods and more vegetables and fruits so you don’t have so many fried foods with so much grease.”* Discuss CATCH with Parents: “*Oh, I would want my parents to know that it’s actually helping everybody and if you want to be healthy you should maybe start it too.”*
Understanding “Go, Slow, Whoa (GSW)”	GSW Meaning: *“Whoa is like really bad for you. Slow is okay to have sometimes and Go is really good for you.”* GSW Messaging Throughout School: “*I seen it in my classroom.”*
Classroom Changes and Habit Changes	Classroom Changes: Energizers and Lessons *“Some of them I like but some of them I don’t because …–it’s not as much movement as it is of some of it.”* “*I like them because like it teaches us things that we can do to keep ourselves healthy….”* *“I don’t like how like you just put up a poster and you don’t hardly talk about it, ….”* Eating Habit Changes: *“...I didn’t like spinach when I was little but then I now learned-when I learned about the “Go, Slow, Whoa” foods I liked it.”*
Changes in Physical Activity	Recess and Physical Education “*They stop letting us have extra, like and we supposed to have at least 40 minutes or 30 minutes of recess. But our class we only have at least 20 minutes to play.”* *“We should play some games. This year we’re actually practicing a lot of sports; he’s teaching us how to do basketball and volleyball. …We’re not playing physical games.”* Fit Fridays *“I like Fit Friday because she gives us examples of things we should eat and how many minutes a day we should exercise and a bunch of stuff.”* *“...well last year my teacher …would wear tennis shoes and he would actually exercise with us and play kickball with us every Friday…”*
**Interviews—Parents/Caregivers of 3rd to 5th Graders**	
**Domains**	**Themes and Illustrative Quotes**
Perception About and Role in CATCH	Mixed Perceptions: “*I believe that the CATCH program is designed to encourage the kids who attend school—and probably the teachers, too—to teach them about the nutrition, to teach them about the importance of exercise, and how to eat and exercise in healthy ways…”* *“I haven’t really heard anything yet.”*
School Changes Related to CATCH	PE and Recess: “*As far as PE, I’ve heard him talk about baseball and that was different.”* *“… at his old school, it was just like you had your basic, old-fashioned jungle gym and that was it. At his school now, they had basketballs out there. Like a lot more equipment type stuff...”* Understanding GSW: “*Those. Yes. She says that’s a whoa food. That’s a go food.”* *“No, [he hasn’t mentioned GSW] but I know that he has a disability, so he may understand it cognitively, but to put it in a way back to you in a way you would understand it, he may not be able to do that.”* Children and Parental Choices *“...I’m a big ranch [dressing] fan…and she made the comment, ‘Well Dad, maybe you should try something else to see what you would like.’”* Class Lessons: *“Well, I know they use it a lot in their math classes and stuff. When they’re converting, they use phrases that were about healthy lifestyles*.” Energizers: *“He does like it. He says that it breaks up the day because my kid is fidgety anyway and he said it breaks up the day and wakes him up and it’s something different…”* Other: *“I know there has been some newer additions to the menu…they have more choices and they will require them to have a fruit with lunch...”*
Opinions on CATCH Changes and Impact on Children/Family	Most Important Changes: “*I think it’s the exercise.”* *“For me personally, I would say her eating habits right now.”* Family Changes: *“…He would not eat squash before. Now he is starting to eat squash.”* *“We are getting at least 30 minutes of exercise a day.”*
Recommendations for CATCH	Expanding CATCH *“I would like to see more brain breaks during the day, maybe less time per brain break to spread them out a little bit.”* *“It’s something that I think needs to be expanded and see if you can get it federally funded.”* Increased Communication with and Participation from Parents. “*Maybe have a CATCH Day, just have something dedicated to that program that allows the parents to really understand it and maybe get involved a little more about it.”*
**Interviews—Staff**	
**Domains**	**Themes and Illustrative Quotes**
Background	“*Actually, I was one of a few people that worked with the principal in researching it and kind of feeling out the staff here to see if they would be interested in implementing the program, and then we brought in the trainers. From there, me, as well as all the other staff members, participated in the formal training.”*
Knowledge of CATCH	*“I think that the program is mainly about educating children on the correct way to take care of themselves to live a healthy lifestyle, which would include making good food choices, and choosing to exercise. That’s probably my best summary of it.”*
Observed Changes at School	Brain Breaks Keep Kids Active and Focused *“...It [brain breaks] definitely keep kids from getting sleepy, or tired, or bored. And I like it myself.”* Changes in Physical Education and Nutrition *“I think that they are more active and they’re constantly participating. There’s not a lot of sit time that they’re waiting for other people to take turns anymore, that the activities that the PE teacher has incorporated has everyone moving all of the time, and there seems to be a variety.”* “*And reading food labels…they really pay attention to that, and they show it to their parents, and they talk about it with them.”* *“It brings that awareness where sometimes it’s hard for, it makes you aware as a teacher too, and as a person. Even in my own life., ‘Oh, what do I need to do?’ because I’m modeling this behavior. Am I making this choice if I’m telling them that?’”*
Views on CATCH Changes	Most Important Changes *“… I feel like the implementing the brain breaks and the more active learning component have been important. I also feel like the nutrition component and the cafeteria, where we have more salads and things like that available, or fresh vegetables and fruit choices than we used to have…the children are making better choices.”*
Recommendations	Staff Want More Collaboration “*Me, I’m really not involved at all…Maybe going into other classrooms to see what they’re doing, so that I can be part of the same thing, collaborating more with each other.”* Staff Want Food Provided to Help Modeling *“I would like to see, maybe like instead of CATCH, just telling us…Why not provide it once a month? The food that we are supposed to—for the kids or even for the teachers.”*

## Data Availability

The data presented in this study are available on request from the corresponding author.

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
