# Peer review of "A Mixed-Method Evaluation of a Rural Elementary School Implementing the Coordinated Approach to Child Health (CATCH) Program"

_nutrients, 2023, doi:10.3390/nu15122729_

Round 1

Reviewer 1 Report

This is a well presented, relevant and timely manuscript. The authors focus on implementation of a multi-method process of addressing childhood obesity within the school environment. I offer a few minor suggestions.

The Introduction is clear and succinct.

Line 110: What is MVP?

The Methods section presents the authors’ appropriate techniques. They outline tools and procedures that resolved discrepancies. I would find it helpful to have “brain break” explained here or in results. GSW was referenced and apparently very effective in recognition and implementation based on comments in the Supplement Section.

The Results are described well with appropriate tables and figures. However, Table 1 lists “Indian” in the “Other” category. Might this be clarified by using “American Indian” or “Native American”? Or, were the participants East Indian? Table 4 alignment was off making it difficult to read on my version.

The Discussion: Could the authors please discuss why the Pacer test results were worse on the posttest than the base line test.

Line 328: Should the word “texts” be “tests”?

Some of the study limitations are acknowledged. Might the authors address the possible roles of genetics and/or natural growth spurts/maturation v. the impact of CATCH in BMI decreases?

Suggestions for future implementation are identified along with the benefits of collaboration and community directed implementation.

Thank you for the opportunity to review this interesting manuscript.

Author Response

Comment 1: What is MVP? (Line 110)

Response 1: Yes, we neglected to define this term. The letters stand for M= Moves and stays active; V= Values healthy eating and mindsets; and P = Practices healthy behaviors.

Revision 1: We’ve updated the text in Materials & Methods (line 110) to include this definition.

Comment 2: I would find it helpful to have “brain break” explained in the Methods or Results section.

Response 2: We agree that this would be helpful and have included a definition in the section where it first appears in the qualitative results text (Line 235).

Revision 2: On line 235, we include this description: (brain breaks or classroom energizers are physical activity breaks used to help kids re-focus and reactivate their brains after they’ve been sitting for extended periods of time)

Comment 3: (a) Table 1 lists “Indian in the “Other” category. Might this be clarified (e.g., American Indian, Native American, or were they East Indian)? (b) Table 4 alignment was off making it difficult to read on my version.

Response 3: (a) The Granville County School District has 2 categories that would include “Indians” – “Asian or Asian Pacific Islanders” and “American Indian or Alaska Natives.” This means Indian could include both groups (where East Indians would fall in the Asian category).

(b) The alignment in Table 4 was likely altered (from left-justified to centered) when the document was converted to a PDF.

Revision 3: (a) Table 1 (line 383) has been modified to include the more detailed group names.

(b) We will use a different approach to converting the tables to a PDF before combining with the manuscript text document. Table 4 as a PDF is also uploaded with this response.

Comment 4: Could the authors please discuss why the PACER test results were worse on the posttest than the baseline test. [Discussion]

Response 4: Because the PACER test data was collected by the school staff, we can only speculate as to why there may have been a lower average score at posttest compared to baseline. The PACER manual lists several possible measurement issues that could impact test scores; we’ve included these as part of our discussion to possibly explain this outcome.

Revision 4: See text inserted on page 7 line 323 of the Discussion.

Comment 5: Should the work “texts” be “tests” (line 328)?

Response 5: Thanks for identifying this typographical error.

Revision 5: The word “texts” has been replaced with “tests.”

Comment 6: Might the authors address the possible roles of genetics and or natural growth spurts/maturation v. the impact of CATCH on BMI decreases? [Study Limitations]

Response 6: There are two things to mention here relative to attributing observed BMI outcomes to the CATCH program. First, this was an evaluation study proposed and funded because CATCH was being implemented in a rural school.  Our study design was specific to the evaluation methodology applied to CATCH as implemented. Since there was no comparison or control school to compare program outcomes, we cannot attribute observed outcomes to the CATCH Program (they could be explained by other community level efforts). Second, regarding the normal changes in BMI for this age group of children, we used changes in BMI z-scores instead of BMI because BMI z-score allowed us to make comparisons between boys and girls and children of different ages, thus accounting for growth patterns and genetic influences.

Revision 6: The study limitations have been revised to expand on the limitations of this evaluation study with mention of the two factors highlighted above. See page 8, line 377.

Reviewer 2 Report

The manuscript, A Mixed-Method Evaluation of a Rural Elementary School Implementing the Coordinated Approaches to Child Health (CATCH) Program, report here the weight status and physical fitness changes after 2 years of CATCH program implementation along with qualitative data describing staff, parents, and student perceptions of the CATCH program. The authors provided detailed procedure of their experiments in the methods section, the experiments were well designed and executed. The overall experimental design is sound and solid, the results are convincing in general. This study opened new windows for future investigations and provided interesting information. The manuscript is qualified to be published in Nutrients, while the following questions are revised.

The title of the paper needs improvement.

The keywords of the paper needs improvement.

Most figures are not labeled with statistical analysis results.

Author Response

Comment 1: The title of the paper needs improvement

Response 1:  We are not quite sure what aspect of the title needs improvement. In selecting the title, we focused on emphasizing that it is the mixed-method evaluation that is the main topic. Our focus is on describing the implementation of an evidence-based intervention (CATCH) in a rural school with racial/ethnic diversity in its student population. The school had already decided to implement the CATCH Program and the academic partners of this rural academic health department secured funds to conduct the evaluation study.

Revision 1: None made because of the reasons listed above.

Comment 2: The keywords need improvement.

Response 2: The approach we used to determine our keywords is that suggested by key resources which suggest choosing words or terms that complement but do not duplicate what’s in the title and abstract. We also aimed to include MeSH (Medical Subject Headings) terms which are often used in search queries. That said, there are several other terms we could have included but did not. We inserted our abstract and introduction into a tool (https://meshb.nlm.nih.gov/MeSHonDemand) that provided additional keywords which we’ve included in the revision. These keywords are also like those in other publications of CATCH.

Revision 2: On page 1, line 44, we have updated our 5 key words and terms to include 3 others: rural populations, physical fitness, and health promotion.

Comment 3:  Most figures are not labelled with statistical analysis results.

Response 3:  We decided to include the statistical analysis results in the text to keep the figures uncluttered, but we can add the values to the figure.

Revision 3: P-values are now added to Figure 1.

Reviewer 3 Report

1. The article provides an evaluation of the implementation of the "CATCH" program for rural children. The "CATCH" program is a very meaningful project.

2. Although the study received ethical approval, there was no information in the article about registration on the clinical research website. Registration on the website is mandatory for clinical research.

3. After 2 years of follow-up, the actual change in BMI from the data was very weak, and no control group was set up. Although the authors refer to this in the discussion, the actual clinical significance is still not negligible.

4. The article does not mention the information on sample size calculation, and it is not clear whether the current sample size meets the minimum sample size requirements, so it is impossible to judge whether the results reflect the real situation.

5. Figure 1 contains too many figures and no figure legend. There are many details in the figure that needs to be carefully adjusted, including inconsistent fonts, text boxes partially obscuring the title(figure 1-4), etc

6. As an intervention study on overweight and obesity, BMI only reflects body shape. More meaningful indicators should be the impact on children's glucose metabolism, lipid metabolism, metabolic syndrome, cardiovascular risk factors, etc. BMI can be used as a primary outcome, but a comprehensive assessment of other metabolic measures may be more valuable for obesity-related studies.

Author Response

Comment 1: [Description of article]

Response 1: None needed.

Revision 1: None needed.

Comment 2:  Although the study received ethical approval, there was no information in the article about registration on the clinical research website. Registration on the website is mandatory for clinical research.

Response 2: To our knowledge, we did not conduct a clinical trial under the NIH definition and the decision tree for NIH clinical trial definition found here: CT-decision-tree.pdf (nih.gov). Because we answered “NO” to “Are participants prospectively assigned to an intervention?” and “Is the study designed to evaluate the effect of the intervention on the participants?” This evaluation study focused more on the implementation of the evidence-based intervention (CATCH) and used a hybrid trial design common in implementation studies (see this reference by Curran et al., 2012 (https://www.ncbi.nlm.nih.gov/pmc/articles/PMC3731143/ ) for 3 types of hybrid designs). As such, this evaluation study using a hybrid implementation study design, and under the specific circumstances of this study (where the decision to implement CATCH happened prior to our evaluation study funding) does not meet the NIH definition of a clinical trial.

Revision 2: None

Comment 3:  After 2 years of follow-up, the actual change in BMI from the data was very weak, and no control group was set up. Although the authors refer to this in the discussion, the actual clinical significance is still not negligible.

Response 3: Though we report 2 years of follow-up, we also note in our description of this CATCH implementation that all the components of the intervention were not implemented, and the school used a sequenced approach because this was what was feasible. Given this context, we would not expect the outcome to represent the maximum effectiveness of the CATCH Program as it was designed. Moreover, without a control group we cannot attribute the outcome in BMI changes to the intervention, though as we explained in our response to Reviewer #1, a decrease in BMI z-score would NOT be consistent with the secular trend in the US. In our manuscript on page 6, lines 302-321, we put our findings in the context of those reported in systematic reviews and noted how they defined “favorable” outcomes for studies without control groups (“studies without a control group that reported a decrease or no change in weight-related outcomes as favorable, regardless of statistical significance, because this showed potential for a decreased rate of change in BMI z-score, overweight, or obesity prevalence.” [29].”).

Revision 3: We refer the reviewer to the discussion on page 6 of our manuscript and the revision to our limitations text to expand on the BMI changes, in response to Reviewer #1 on page 8, lines 377-382.

Comment 4: The article does not mention the information on sample size calculation, and it is not clear whether the current sample size meets the minimum sample size requirements, so it is impossible to judge whether the results reflect the real situation.

Response 4: As this was an implementation hybrid trial and not a clinical trial designed to test differences between groups, we would not have done a sample size calculation. If, however, we wanted to estimate the minimum total sample size for a two-tailed t-test study, we could use an a-priori sample size calculator to get an estimate (using anticipated effect size (Cohen’s d), statistical power, and alpha level). For a medium effect size (0.5) with 80% power and alpha = .05 the minimum total sample size for a 2-tailed hypothesis yields a sample size of 128. For a small effect size (0.2) the minimum total sample would be n= 788. With our sample of 157, and assuming a 0.59 standard deviation in mean BMI z-score which we observed in our overall sample, with an anticipated medium effect size of 0.45, we would have 80% power to detect a 0.26 difference in BMI z-scores between groups. For an anticipated small effect size of 0.2, a sample of 157 would give us 80% power to detect a between group difference of 0.12, which is greater than what we observed in our single group. Sample size aside, our outcomes were similar to the median change in BMI z-scores observed in 10 studies from the 2023 systematic review of school dietary and physical activity interventions (ref. #29).

Revision 4: No revisions made given our response above.

Comment 5: Figure 1 contains too many figures and no figure legend. There are many details in the figure that needs to be carefully adjusted, including inconsistent fonts, text boxes partially obscuring the title (figure 1-4), etc.

Response 5: We agree that there are too many figures and note that in the combining of files for submission some formatting was lost.

Revision 5: We have separated the single figure into 3 figures. Figure 1 now includes overall and sub-group comparisons in BMI z-score; Figure 2 shows BMI percentile data with sub-group analyses, and Figure 3 includes the physical activity outcomes. We have also cleaned up the formatting by creating PDFs prior to collating documents. The relevant text on page 6 has also been updated.

Comment 6: As an intervention study on overweight and obesity, BMI only reflects body shape. More meaningful indicators should be the impact on children's glucose metabolism, lipid metabolism, metabolic syndrome, cardiovascular risk factors, etc. BMI can be used as a primary outcome, but a comprehensive assessment of other metabolic measures may be more valuable for obesity-related studies.

Response 6: These are all valid comments for a clinical trial to determine the effectiveness of interventions like CATCH. But CATCH has already been determined to be effective in larger cluster RCTs and this study is not such as study. As mentioned earlier and in our manuscript, this is more of an implementation study that involved academic and community partners in describing how CATCH was implemented, what students, parents, and school staff thought of the program, and the program outcomes using data that is easily collected, affordable/feasible given our funding, and acceptable for school-based research in this small rural community. Many studies evaluating CATCH have used BMI z-scores to assess changes in BMI rather than BMI itself because it allows you to compare boys vs. girls and children of different ages.  BMI z-scores also let you identify whether there are improvements over time relative to reference values.

Revision 6: Given our response, we do not think revisions are needed.

Round 2

Reviewer 3 Report

The format of Figure 5(PACER Test)needs to be checked